# Impact of Different Resistance Training Protocols on Balance, Quality of Life and Physical Activity Level of Older Women

**DOI:** 10.3390/ijerph191811765

**Published:** 2022-09-18

**Authors:** Luis Leitão, Gabriela R. O. Venturini, Ricardo Pace Junior, Estêvão Rios Monteiro, Luiz Guilherme Telles, Gleisson Araújo, Jefferson Novaes, Carlos Tavares, Sílvio Marques-Neto, Mauro Mazini

**Affiliations:** 1Sciences and Technology Department, Superior School of Education, Polytechnic Institute of Setubal, 2910-761 Setúbal, Portugal; 2Life Quality Research Centre (CIEQV), 2400-901 Leiria, Portugal; 3Laboratory of Physical Activity and Health Promotion, State University of Rio de Janeiro (UERJ), Rio de Janeiro 20550-900, Brazil; 4Federal Center of Technological Education of Minas Gerais, Leopoldina 36700-000, Brazil; 5Graduate Program of Physical Education of Fasar, Santa Rita Faculty, Conselheiro Lafaiete 36400-000, Brazil; 6Postgraduate Program in Physical Education, Federal University of Rio de Janeiro, Rio de Janeiro 21941-901, Brazil; 7Undergraduate Program in Physical Education, Augusto Motta University Center, Rio de Janeiro 21041-020, Brazil; 8Undergraduate Program in Physical Education, IBMR University Center, Rio de Janeiro 22631-002, Brazil; 9Graduate Program in Physical Education, Sudamerica Faculty, Cataguases 36774-552, Brazil; 10Physical Activity Sciences Graduate Program, Salgado de Oliveira University (UNIVERSO), Rio de Janeiro 24030-060, Brazil; 11Physical Education Graduate School, Estácio de Sá University (UNESA), Rio de Janeiro 20771-004, Brazil

**Keywords:** older women, resistance training, quality of life, physical activity level, balance

## Abstract

Background: Physical activity (PA) and physical fitness are key factors for quality of life (QoL) for older women. The aging process promotes the decrease in some capacities such as strength, which affect the activities of daily life. This loss of strength leads to a reduction in balance and an increased risk of falls as well as a sedentary lifestyle. Resistance Training (RT) is an effective method to improve balance and strength but different RT protocols can promote different responses. Power training has a higher impact on the performance of activities of daily life. Therefore, our study aimed to analyze if different RT protocols promote individual responses in balance, QoL and PA levels of older women and which are more effective for the older women. Methods: Ninety-four older women were divided into four RT groups (relative strength endurance training, SET; Traditional strength training, TRT; absolute strength training, AST; power training, PWT) and one control group (CG). Each RT group performed a specific protocol for 16 weeks. At baseline and after 8 and 16 weeks, we assessed balance through the Berg balance scale; PA levels with a modified Baecke questionnaire and QoL with World Health Organization Quality of Life—BREF (WHOQOL-BREF) and World Health Organization Quality of Life—OLD module (WHOQOL-OLD). Results: Balance improved after 16 weeks (baseline vs. 16 weeks; *p* < 0.05) without differences between all RT groups. PWT (2.82%) and TRT (3.48%) improved balance in the first 8 weeks (baseline vs. 8 weeks; *p* < 0.05). PA levels increased in PWT, TRT and AST after 16 weeks (baseline vs. 16 weeks; *p* < 0.05). Conclusion: All RT protocols improved PA levels and QoL after 16 weeks of training. For the improvement of balance, QoL and PA, older women can be subjected to PWT, AST and SET, and not be restricted to TRT.

## 1. Introduction

Worldwide, the population over the age of 60 years old and their average life expectancy is increasing year after year, mainly due to improvements in the health system and new technologies [1,2]. The aging process is associated with changes in older women affecting their daily routine, quality of life (QoL) and health. Sarcopenia is a syndrome that occurs after 60 years and is characterized by a progressive loss of muscle mass and strength that are associated with declines in functional capacity and loss of balance, and increases in the risk of falling [3,4,5].

Several worldwide organizations recommended resistance strength (RT), endurance training, interval training, aerobic training and multicomponent training executed at moderate-to-vigorous intensities, as methods to improve functional capacity and QoL [6,7,8]. RT is an effective method to increase fat-free mass, bone strength, muscle strength and functional capacity, and reduce risk factors for several diseases such as cardiovascular and metabolic diseases [3,4,6,7,9]. Filho et al. [10] reported that older women, after twenty weeks of strength training, improved functional capacity, strength and the ability to perform activities of daily life (ADL). In addition to these effects, RT increased psychosocial health, which combined, has an important influence on QoL. Additionally, physical fitness and physical activity (PA) combined can have considerable consequences on the performance of ADL and QoL [6,11]. For older women, QoL is becoming one of the most important factors as they become older since the extended years must be followed up with a healthy life [4,12]. Puciato et al. [13] showed that higher PA levels resulted in higher physical, psychosociological and environmental QoL domains.

Several studies [6,11,14,15,16] have focused on the impact of endurance-based exercise and strength training on balance and QoL. Fragala et al. [7] suggested that RT is crucial for improving and maintaining muscle strength, psychological well-being, QoL and healthy life expectancy. The size of those benefits is dependent on the type of RT protocol used. Traditional resistance training (TRT) is performed 2–3 times a week based on relative strength with 8–12 repetitions at 60–80% of 1 RM. Strength endurance training (SET) comprises a higher number of repetitions at a light-to-moderate intensity until fatigue. Absolute strength training (AST) consists of no more than six repetitions at a higher amount of strength than the neuromuscular system performs [7,10]. Power training (PWT) is characterized by performing the concentric phase of the movement at the maximum amount of force as fast as possible with loads of no more than 60% of 1 RM. According to different studies, power training (PWT) is important for functional improvements in ADL and balance and can promote more power at lower external resistances, which will result in a higher velocity of power to perform any task [6,7,10,17].

All these different RT types are important for the balance and QoL of older women. According to the literature, it is not clear if any RT protocol can promote higher improvements in balance when compared to the other protocols. Diverse prescriptions of RT can promote different dose–response relationships since intensity and volume are key factors for positive adaptations [7]. To our knowledge, there is no study in the literature that compared all these four protocols (TRT, SET, PWT and AST) in the same investigation at the same time. Thus, we propose to analyze the effects of 16 weeks of four different RT protocols on QoL, PA levels and balance of older women and see if there are any individual responses from each one.

## 2. Materials and Methods

### 2.1. Experimental Design

This was a sixteen-week study that consisted of assessing the effects of TRT, SET, PWT and AST on balance, PA levels and QoL in older adults. To analyze these effects, all participants visited the laboratory at three specific times. The first visit was after the first two weeks of familiarization with RT, and then after 6 weeks of adaptation to RT, and the third assessment was performed 24 h after the last training session after eight weeks of specific RT. During all three visits, the participants performed the Berg Balance Scale and completed a questionnaire which comprised the modified Baecke questionnaire, World Health Organization Quality of Life—BREF (WHOQOL-BREF) and World Health Organization Quality of Life—OLD module (WHOQOL-OLD). All assessments were conducted in the same location and by experienced instructors in older adult training.

### 2.2. Sample and Ethical Procedures

The inclusion criteria were (a) being aged between 60–75 years old; (b) the ability to perform exercise without contraindication and attendance of at least 85% of the sessions. Exclusion criteria were (a) any osteoarticular problem that affects the performance of any exercise; (b) medical contraindications (e.g., heart problems, surgeries) and (c) involvement at the same time in any PA program. All procedures of the study were accomplished in accordance with the Declaration of Helsinki and approved for human experiments by local institutional ethical committee (protocol number 2.887.652). Before studying experimental protocols, all volunteers signed an informed consent form. Of one hundred fifty-six volunteers only ninety-five older women fulfilled the inclusion criteria and were randomly assigned to 5 groups: SET, AST, PWT, TRT and control group (CG) (Table 1). All participants were instructed to keep, over the study, their normal lifestyle and were advised not to consume coffee, tea, alcohol or tobacco and do any vigorous exercise during the last 24 h before all assessments. The volunteers of the CG did not perform any systematic PA during the 16 weeks and maintained their daily routine. During the study, five older women dropped out (one from CG and TRT; three from PWT) due to illness (3) or not attending at least 85% of the sessions (2).

#### 2.2.1. Resistance Training Protocol

The RT protocol consisted of training sessions twice a week with 72 h to 96 h of recovery between sessions for 16 weeks. The first two weeks for familiarization are followed by six weeks of adaptation and eight weeks of specific RT. All groups performed the same exercises (curl-ups, seated row, horizontal leg press, machine bench press, leg extension, and seated leg curl) in the same order, volume of repetitions and recovery interval. To assess muscle strength, 10 RM test was used [10]. Load progression was 5 to 10% every two weeks and rating of perceived exertion (RPE) [18] was used for load control. In familiarization and adaptation period, we used values of RPE from 5 to 7, and for specific RT period, we used RPE values from 6 to 8.

In the familiarization period, all volunteers performed 2 sets of 15 repetitions at 50–60% of 10 RM with one minute of rest between series and exercises. The six weeks of adaptation were performed with 3 sets of 12–15 repetitions at 60% of 10 RM with one minute of rest between sets and two minutes between exercises. During the last eight weeks of RT, all groups performed their specific training with training load adjusted using RPE: SET—1 set of 20–25 repetitions; AST—4–5 sets of 4–5 repetitions; TRT—2–3 sets of 8–12 repetitions; PWT—2–3 sets of 8–12 repetitions. To equalize the total number of repetitions, all groups performed 20–25 repetitions in every exercise with three minutes of rest between sets and exercises. The concentric and eccentric phase lasted two seconds, controlled by a metronome, for TRT, AST and SET. For PWT, the concentric phase was performed at maximum velocity (50% of 10 RM). No participant performed any repetitions until failure [19].

#### 2.2.2. Anthropometric Measures and Balance

Body mass (kg) and body mass index (BMI, kg·m^−2^) were measured through an Omron Scale (OMRON, Kyoto, Japan), and height was assessed with a stadiometer (Sanny, Fortaleza, Brazil), all in the first visit.

The balance of the older women was assessed from the performance of fourteen items scored from 0 to 4 (0—unable to perform; to 4—task complete with full independence) on the Berg Balance Scale: Sitting to standing; stand up for 2 min without support; stand up with eyes closed; stand up with feet together; standing to sitting; sitting with back unsupported; transferring with and without support; reaching forward with outstretched arms; pick up objects from the floor; turning to look behind oneself; turning 360°; placing one foot on a step; placing one foot in front of the other; and stand up on one leg. This scale allowed us to measure the risk of falls and quantitatively the balance of older women [20,21].

#### 2.2.3. Level of Physical Activity and Quality of Life

To measure PA levels we used modified Baecke questionnaire (MBQ) [22] and for QoL we used WHOQOL-BREF and WHOQOL-OLD [23].

The MBQ allowed us to measure habitual PA of the older women through the score of three specific domains: work activity; sports activity; and leisure activity. The WHOQOL-BREF is an instrument that is frequently used to assess QoL of healthy and ill populations. It comprised twenty-four questions covering four domains (physical, psychological, social relationships and environment) and two global questions of QoL in a total of twenty-six questions. WHOQOL-OLD questionnaire is an add-on module from WHOQOL-BREEF specific to older adults, with 24 items categorized into six facets: sensory abilities; autonomy; past, present and future activities; social participation; death and dying; and intimacy.

### 2.3. Statistical Analysis

Descriptive statistics were presented by mean ± standard deviation. Levene, Mauchly and Shapiro–Wilk tests were used to verify the assumptions of the sphericity, variance homogeneity and normality of the data, respectively. One-factor ANOVA was used to analyze sample characteristics and two-way ANOVA was performed to compare groups during the training period. Bonferroni adjustment was used for multiple comparisons. For relative differences we used delta percentage (∆% = [(post-test score − pretest score)/pretest score] × 100). Statistical significance was maintained at 5%. For data analysis, we used SPSS v.23. Previously, a sample size calculation was performed with G*Power 3.1 (ver. 3.1.9.7; Heinrich-Heine-Universität Düsseldorf, Düsseldorf, Germany) with a power of 0.8, and a total of 100 older women was necessary, 20 in each group.

## 3. Results

The participants’ attendance rate was 87%. All RT protocols resulted in improvements in balance (Figure 1 and Table 2), total PA and QoL (Table 3).

For balance, we verified that after 16 weeks, all RT groups improved but only SET (*p* = 0.046) and TRT (*p* = 0.036) improved after the first 8 weeks. All RT groups finished the study with better results than the CG (*p* < 0.05).

PA levels increased after 16 weeks in all RT groups, despite the baseline values already being high (*p* < 0.05), except for SET (Table 3). With the questionnaire of WHOQOL-BREF, we reported no changes in the psychological domain. In the social domain, PWT (−6.5%) and TRT (−6.5%) decreased (*p* < 0.05). The physical domain of QoL only improved in SET (3.9%) after 16 weeks (*p* < 0.05). With the specific questionnaire for older adults, WHOQOL-OLD, 16 weeks resulted in improvements in QoL for SET (7.9%), PWT (17.5%), TRT (17.2%), and AST (14.5%).

## 4. Discussion

Balance in older women is crucial to reduce the risk of falling and is a key component of ALD. The purpose of this study was to analyze the possible individual dose–response of four different types of RT on the balance, PA and QoL of older women. Our results showed that after 16 weeks, (a) QoL and PA levels increased in all RT groups with higher improvements in TRT and PWT; (b) balance improved which reduced the risk of falling; and (c) every RT protocol resulted in similar results in balance despite their prescription specifications, which allows for concluding that balance reacts similarly with all of them.

All RT protocols promoted similar benefits in balance and reduced the risk of falling. The improvement of balance may result from the increased lower body strength and muscle mass after 16 weeks. A stronger lower body leads to a more stable base of support and a reduced risk of falling [24]. Additionally, all RT can promote a higher bone density, better metabolic capacity of skeletal muscle and higher gait speed that contributed to higher values of balance [7]. According to different studies [11,16,25,26], one huge problem is the risk of falling. Older women who fall can develop a fear of falling that will limit their PA and mobility. After falling, the risk of institutionalization increases and QoL is reduced due to loss of autonomy and mobility and will lead to lower PA levels. Associated with balance is also cognition, which regulates and controls mobility and can increase the level of PA and functional capacity. Both capacities are crucial to perform ADL and for improving QoL in older women [4,11,22,27]. Our results showed that the increase in balance in all RT groups promoted higher levels of QoL and PA levels. Additionally, Sampaio et al. [4] concluded that higher levels of balance are associated with general cognition.

Increasing PA can reduce the risk of mortality and some risk factors for cardiovascular diseases, and in addition, can improve aspects related to QoL. In our study, we observed that total PA increased in PWT, TRT and AST, which can lead to the observed increase in QoL because the highest levels of PA and physical fitness have a huge impact on QoL [5]. Another result we observed in all RT groups was that leisure and locomotion PA increased. According to Scarabottolo et al. [26], this result is positively associated with mental health and total PA levels [28,29], and can also explain the interest of the participants. Deciding which activity they are likely to perform will increase central and autonomous motivation [30].

The improvement of QoL observed in all RT groups from WHOQOL-BREF [5,13] can be explained in part by the higher values of all PA dimensions because enough energy, pain freedom and great ability to perform ALD are fundamental factors for QoL [31]. The environmental domain was the domain that had the lower value of QoL, and the physical and psychological domains were higher. This result is in accordance with Wong et al. [32] and can be explained through the moderate/vigorous intensity used in all RT protocols that correlate with psychological health [33] and higher QoL levels [15,16].

In the WHOQOL-OLD questionnaire for QoL, we observed higher values in all RT groups after 16 weeks. According to Haider et al. [25], balance can justify our results because it is the strongest factor associated with autonomy, physical health and social participation. Additionally, balance is associated with mental health and well-being [11], factors that are very important for social participation [25,34]. Kwon et al. [35], after 12 weeks of exercise, reported that QoL improved but not physical performance. Social participation has a crucial role in producing positive effects on the well-being and QoL of older adults [35,36,37], and not only the physical benefits that resulted from our program. Therefore, exercise promotes positive perceptions of QoL and not just physical and mental benefits [5,38,39], regardless of the type of RT.

All four RT protocols resulted in benefits for older women without differences between the groups after 16 weeks. Some possible reasons could be the higher baseline values that allowed our older women higher autonomy and independence for similar outcomes for all protocols. Balance values in all groups before the RT protocol were near the maximum of each BBS test, which may have promoted a similar adaptation. Additionally, good mobility and perception of health and emotional status observed in QoL and PA questionnaires were determinants of our results [40]. Toselli et al. [12], with institutionalized older adults, reported that low levels of PA, autonomy and mobility promoted low levels of QoL, and interventions of active aging should be engaged. QoL should be prioritized in later life in preference to disease-based outcomes because older women give more importance to the way they see themselves [33]. RT is crucial to improve QoL in the later years of older women [11,27].

Our study has some limitations: (a) not controlling dietary routine can interfere with the performance of the older women during RT sessions; (b) the age range of all participants limits our results to be used with older adults of more than 75 years; and (c) our participants already had higher baseline values. In this way, for future studies, we recommend more research on the impact of different RT in older women with low levels of balance and PA, and with different age groups.

## 5. Conclusions

After 16 weeks of SET, PWT, TRT and AST, the older women improved their balance, QoL and PA levels. Different RT protocols promoted similar responses, reinforcing the importance of all types of RT as a crucial method for health promotion. Although PWT and TRT seem to produce more effects in QoL for exercise professionals, we recommend all RT protocols to increase/maintain QoL and balance in older women and not only TRT. Moreover, associating the personal interests of older women with RT protocol selection can increase the final results.

## Figures and Tables

**Figure 1 ijerph-19-11765-f001:**
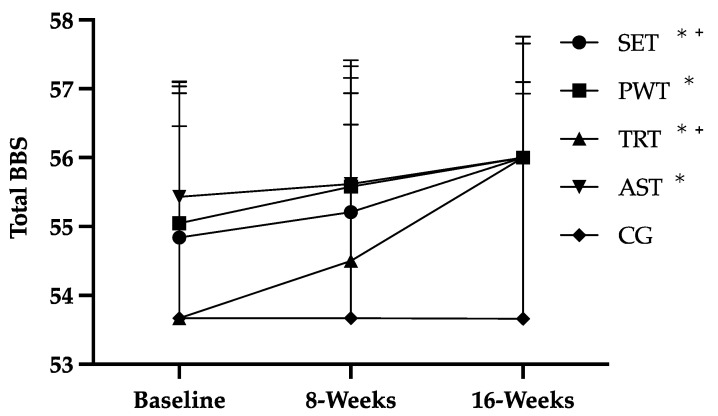
Effects of RT on balance of older women; traditional resistance training (TRT); strength endurance training (SET); power training (PWT); and absolute strength training (AST); total Berg balance scale score (Total BBS); * *p* < 0.05, 16 weeks vs. Baseline; ^+^
*p* < 0.05, 8 weeks vs. Baseline.

**Table 1 ijerph-19-11765-t001:** Participants at baseline (mean ± SD).

Variable	SET (*n* = 20)	PWT (*n* = 19)	TRT (*n* = 17)	AST (*n* = 20)	CG (*n* = 19)
Age (years)	64.1 ± 4.10	66.21 ± 3.10	67.21 ± 5.10	67.12 ± 5.81	69.03 ± 3.42
Height (cm)	154.38 ± 5.24	158.21 ± 4.55	154.79 ± 7.55	157.86 ± 5.64	156.24 ± 6.30
Body weight (kg)	69.10 ± 14.8	71.90 ± 16.03	61.40 ± 10.51	75.61 ± 67.2	67.21 ± 11.46
BMI (kg/m^2^)	29.03 ± 5.53	28.73 ± 5.74	25.74 ± 4.64	30.72 ± 5.74	27.42 ± 5.41

Traditional resistance training (TRT); Strength endurance training (SET); Power training (PWT); Absolute strength training (AST); Body Mass Index (BMI).

**Table 2 ijerph-19-11765-t002:** Berg Balance Scale.

	Baseline	8 Week	16 Week	Baseline vs. 16-Week *p*-Value
SET	54.57 ± 2.32	55.05 ± 2.12 ^+^	55.89 ± 0.32 *	0.02
PWT	54.95 ± 1.96	55.47 ± 1.84	55.95 ± 0.23 *	0.04
TRT	53.55 ± 3.38	54.50 ± 2.66 ^+^	55.89 ± 0.47 *	0.01
AST	55.43 ± 1.03	55.62 ± 0.87	55.95 ± 0.22 *	0.02
CG	53.44 ± 3.28	53.28 ± 3.39	53.56 ± 3.18	0.43

Traditional resistance training (TRT); strength endurance training (SET); power training (PWT); and absolute strength training (AST); * *p* < 0.05, 16 weeks vs. Baseline; ^+^
*p* < 0.05, 8 weeks vs. Baseline.

**Table 3 ijerph-19-11765-t003:** RT-type effects in PA and QoL.

		Group	Baseline	8 Week	16 Week	Baseline vs. 16-Week *p*-Value
PA Levels	Occupational PA	SET	2.11 ± 0.32	2.17 ± 0.37	2.25 ± 0.41	0.25
		PWT	1.97 ± 0.32	2.03 ± 0.34	2.04 ± 0.36	0.08
		TRT	1.97 ± 0.25	1.99 ± 0.32	2.07 ± 0.27	0.09
		AST	2.05 ± 0.35	2.11 ± 0.4	2.14 ± 0.38	0.51
		CG	1.88 ± 0.12	1.91 ± 0.38	1.89 ± 0.38	0.91
	Leisure Sports Practice	SET	2.21 ± 0.16	2.22 ± 1.18	2.2 ± 1.15	0.33
		PWT	2.03 ± 0.80	2.12 ± 0.78	2.12 ± 1.78	0.07
		TRT	2.11 ± 0.96	2.87 ± 1.96	2.81 ± 1.90 *	0.04
		AST	1.83 ± 0.12	1.91 ± 0.78	2.4 ± 0.21 *	0.03
		CG	1.46 ± 0.68	1.43 ± 0.68	1.38 ± 1.62	0.85
	Leisure and locomotion PA	SET	2.99 ± 1.82	3.43 ± 1.89	3.06 ± 1.15	0.07
		PWT	1.36 ± 0.91	2.18 ± 1.49	2.47 ± 1.37 *	0.05
		TRT	1.27 ± 0.57	1.82 ± 0.95	1.81 ± 0.95 *	0.02
		AST	1.69 ± 0.82	1.91 ± 0.58	2.21 ± 0.69 *	0.02
		CG	1.41 ± 0.31	1.52 ± 0.6	1.43 ± 0.41	0.97
	Total PA	SET	7.31 ± 1.73	7.82 ± 1.95	7.51 ± 1.41	0.31
		PWT	5.36 ± 1.02	6.33 ± 1.53	6.63 ± 1.62 *	0.04
		TRT	5.35 ± 1.24	6.68 ± 1.30	6.69 ± 1.13 *	0.05
		AST	5.57 ± 1.37	5.93 ± 1.91	6.75 ± 1.79 *	0.04
		CG	4.75 ± 1.71	4.86 ± 1.52	4.71 ± 1.06	0.92
WHOQOL-BREF	Physical	SET	76.32 ± 15.02	77.26 ± 16.16	79.32 ± 15.04 *	0.05
		PWT	74.01 ± 16.88	75.20 ± 19.31	73.21 ± 18.34	0.86
		TRT	72.06 ± 12.71	76.72 ± 14.89 *	74.58 ± 12.62	0.71
		AST	71.99 ± 15.89	76.13 ± 15.48 *	73.12 ± 16.17	0.51
		CG	72.56 ± 15.62	72.37 ± 16.49	73.50 ± 15.85	0.98
	Psychological	SET	78.29 ± 14.27	80.04 ± 11.69	78.29 ± 10.72	0.94
		PWT	78.47 ± 15.41	79.40 ± 15.82	73.38 ± 14.09	0.31
		TRT	75.98 ± 15.13	75.00 ± 16.21	73.78 ± 14.19	0.46
		AST	78.51 ± 11.64	76.97 ± 15.30	76.10 ± 11.85	0.51
		CG	79.39 ± 12.69	82.68 ± 12.75	80.26 ± 11.69	0.74
	Social	SET	77.19 ± 16.16	75.88 ± 18.40	78.07 ± 15.27	0.83
		PWT	82.41 ± 12.75	79.17 ± 11.52 *	76.85 ± 15.54 *	0.04
		TRT	75.49 ± 16.53	72.06 ± 18.85 *	66.18 ± 21.14 *	0.02
		AST	78.51 ± 17.42	78.51 ± 17.19	75.88 ± 15.19	0.41
		CG	77.19 ± 14.12	79.39 ± 14.26	78.07 ± 13.38	0.86
	Environment	SET	71.38 ± 21.93	74.34 ± 18.30	72.86 ± 12.42	0.77
		PWT	65.97 ± 13.55	68.75 ± 17.58	68.58 ± 16.66	0.57
		TRT	66.54 ± 17.04	67.83 ± 15.18	66.18 ± 13.22	0.96
		AST	71.22 ± 16.91	70.39 ± 15.81	65.63 ± 14.47 *	0.05
		CG	74.84 ± 13.20	73.85 ± 13.27	74.01 ± 14.06	0.84
	Total WHOQOL-BREF	SET	75.76 ± 13.98	77.02 ± 14.05	77.18 ± 9.09	0.61
		PWT	73.88 ± 12.27	75.11 ± 14.16	72.70 ± 13.87	0.76
		TRT	72.18 ± 10.96	73.14 ± 12.14	71.32 ± 8.33	0.71
		AST	74.55 ± 10.98	75.30 ± 12.28	72.12 ± 10.20	0.41
		CG	75.46 ± 11.32	76.16 ± 11.62	75.71 ± 11.14	0.92
WHOQOL-OLD	Total WHOQOL-OLD	SET	54.66 ± 11.18	55.76 ± 11.60	58.96 ± 10.56 *	0.02
		PWT	56.91 ± 10.03	57.89 ± 11.09	66.83 ± 9.42 *	0.01
		TRT	52.14 ± 11.54	55.39 ± 14.43	61.15 ± 11.62 *	0.02
		AST	57.55 ± 9.83	63.80 ± 12.62 *	65.89 ± 10.70 *	0.05
		CG	62.05 ± 10.41	61.97 ± 10.72	62.16 ± 10.73	0.89

Traditional resistance training (TRT); strength endurance training (SET); power training (PWT); and absolute strength training (AST); * *p* < 0.05 vs. Baseline.

## Data Availability

Available through the corresponding author by request.

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
