# Peer review of "Impact of Different Resistance Training Protocols on Balance, Quality of Life and Physical Activity Level of Older Women"

_ijerph, 2022, doi:10.3390/ijerph191811765_

Round 1

Reviewer 1 Report (Previous Reviewer 1)

Thank you for making the requested edits to your manuscript. I believe this article is ready to move towards publication.

Author Response

Dear Reviewer,

We are grateful for your consideration on our manuscript, and we also very much appreciate your last suggestions which allowed us to improve our article.

Reviewer 2 Report (New Reviewer)

General comments:

Authors report the effects of 16 weeks of resistance training interventions versus a control condition on balance, physical activity, and quality of life in 94 older women who were randomized into one of five (5) groups [relative strength endurance training (SET), Traditional strength training (TRT), absolute strength training (AST), power training (PWT) and one control group (CG)]. Generally, the authors observed positive effects of the different resistance protocols on physical activity and quality of life after the 16-week intervention. They also report an improvement in balance function after the interventions without group differences. This is a relevant study given the role of physical exercise in promoting healthy ageing while retarding ageing-related musculoskeletal functional decline and consequences such as the increased risk of falls. The manuscript is well-written, however, it would benefit from English proofreading.

ABSTRACT:

·      Please specify the time points during which assessments of outcome variables were done.

·      Lines 39-40: for clarity, specify the reference time point(s) for comparisons.

·      Line 37: “WHOQOL”, “BREF” and “WHOQOL-OLD” should be fully defined.

·      Line 42: replace “submitted with “subjected” or “exposed”.

INTRODUCTION:

·       Line 87: indicate the four different protocols.

METHODS:

·      While it’s implied, please insert a description of treatment for the control group.

·      Lines 168-169: statistical analysis – these lines should be re-written for clarity.

·      Please specify if the sample size was calculated a priori or post-hoc.

RESULTS:

·      Good!

DISCUSSION:

·      Line 219: Is it “both” or “booth”? Please correct it.

Author Response

Dear Reviewer,

We are grateful for your consideration on our manuscript, and we also very much appreciate your last suggestions. All your suggestions were attended and are highlighted in yellow.

ABSTRACT:

  • Please specify the time points during which assessments of outcome variables were done.
  • Lines 39-40: for clarity, specify the reference time point(s) for comparisons.
  • Line 37: “WHOQOL”, “BREF” and “WHOQOL-OLD” should be fully defined.
  • Line 42: replace “submitted with “subjected” or “exposed”.

A: All abstract suggestions were attended.

INTRODUCTION:

  • Line 87: indicate the four different protocols.

A: We add in the text.

METHODS: 

  • While it’s implied, please insert a description of treatment for the control group.
  • Lines 168-169: statistical analysis – these lines should be re-written for clarity.
  • Please specify if the sample size was calculated a priori or post-hoc.

A: All suggestions were attended.

DISCUSSION: 

  • Line 219: Is it “both” or “booth”? Please correct it.

A: We changed.

This manuscript is a resubmission of an earlier submission. The following is a list of the peer review reports and author responses from that submission.

Round 1

Reviewer 1 Report

Thank you for making revisions to your manuscript entitled “Impact of different Resistance Training protocols on Balance, quality of life and physical activity level of older women.” A few areas still remain to be addressed before this manuscript can move towards publication.

Abstract:

1.       The abstract explains the background of why resistance training is important for this population but does not identify a gap in the literature that needs to be filled by this study. Is it that we are unsure which RT type best promote the outcomes of focus?

Introduction:

2.       The introduction does a nice job summarizing some of the background but it does not ID where there is a gap in knowledge. Why is the RT 16 weeks? Is that a unique length of time there is not a lot of data to support? Why the particular selection of the 4 RT types? As stated above is the gap which is most advantageous for the outcomes of focus?

Results:

3.     Table 2: it would be helpful to add a column next to 16 week column providing the p-value to see how close some changes came to significance and to determine those that are identified as significant (*) were beyond the 0.05 criteria.

Discussion/Conclusion:

4.     Why does the last sentence of the first discussion paragraph only state that similar improvement were made following all RT but only mentions balance. Later on in the discussion it appears that PA and QoL have divergent findings based on RT group. These RT specific changes should be highlighted in the first paragraph of the discussion.

5.     In the discussion, specify what values at baseline were higher and how each of these elevated outcomes may have influenced your results.

Reviewer 2 Report

INTRODUCTION

Page 3, Lines 99-107 – References are needed for all this types of training.

METHOD:

Page 3 – Lines 132-136 – Who examined the participants and confirmed that they are eligible for the program?

Page 3, Line 143 – replace with “…not to consume coffee…”

What is the average age of participants? How many participants were in each group? This needs to be highlighted in the text also.

The names of the exercises you are mentioning are not usual and it is not clear what exactly did they perform. Please change the names or put explanations.

You used 20 RM test to assess muscle strength. Please explain it in more details.

Page 4 – Line 164-166 – This sentence is unclear. Which exercise they perform 20-25 repetitions? In which group?

The amount of load for each group is not equal. It is not the same thing to perform 20-25 repetitions in the muscle endurance and absolute strength training. Also, if you want to hit specific goals, you need to adjust RM% for each person and for each exercise. From your text this looks pretty confusing and not appropriate.

Page 4 – Line 178 – 180  -This test needs to be explained.

Page 4 – Line 182 – 190 – More details about the questionnaires should be provided.

Which software did you used for calculations?

RESULTS:

Table 2 is confusing, the results should be presented more clearer.

DISCUSSION:

It is not a surprise that a group of older women had improvements in PA and QOL after nearly 4 months of regular exercise.

You wrote “All four RT protocols resulted in benefits for the older women without differences between groups after 16 weeks.” – This is somewhere expected since they perform same number of repetitions and used same RM%. This is a clear limitation of the study.

You did not explained the physiological background of increased balance. This needs to be explained in detail.

CONCLUSION:

Directions for future studies should be added.

Reviewer 3 Report

Dear corresponding Author, thanks for submitting your paper.

I appreciated the idea of your research, it could have many application for professionals. At the same time I want to underline some particular aspects that compromise your paper.

1) In the introduction you describe the different RT protocols. The description is well done but it lacks of literature support. Especially for the training load parameters. It is not clear if the different protocols are divided by yourself or if you took these information from the literature.

2) In the method, particularly in 2.2.1 paragraph, you describe the process from familiarization to specific training. You declare the load % for the familiarization but not for specific training, or in case you did it is not clear enough.

3) Figure 1 and results reporting. In Figure 1 it is clear that the baseline values of BBS are different between CG and the other groups. It could be a problem in terms of successive modification, moreover no data and no number are detailed provided for the BBS. Other two aspects are very curious in this figure: 1) the CG is exactely always the same. Yes it is a control gruop, ok, but this behavior is suspected. 2) The final score at the end of the 16-weeks is exactely the same for any RT protocol. It is suspected as well. No comments are provided about this neither in the results nor in the discussion.